# Multiple-Vessel-Based Blood Gas Profiles Analysis Revealed the Potential of Blood Oxygen in Mammary Vein as Indicator of Mammary Gland Health Risk of High-Yielding Dairy Cows

**DOI:** 10.3390/ani12121484

**Published:** 2022-06-08

**Authors:** Juan Feng, Wenchao Peng, Zhenzhen Hu, Jie Cai, Jianxin Liu, Diming Wang

**Affiliations:** MoE Key Laboratory of Molecular Animal Nutrition, Institute of Dairy Science, College of Animal Sciences, Zhejiang University, Hangzhou 310058, China; 22017005@zju.edu.cn (J.F.); 21817076@zju.edu.cn (W.P.); 12117005@zju.edu.cn (Z.H.); zjcaijie@zju.edu.cn (J.C.); liujx@zju.edu.cn (J.L.)

**Keywords:** high-yielding dairy cows, blood gas profiles, oxygen, somatic cell count, mammary gland health

## Abstract

**Simple Summary:**

The objective of the study was to realize the function of blood gas parameter in relation to the health of high-yielding dairy cows. The study showed that most blood gas parameters differ across the three vessels (coccygeal artery, coccygeal vein, and mammary vein). The associations of oxygen-related variables in the mammary vein are negatively correlated with the levels of malondialdehyde, lactate dehydrogenase, and plasmin in the milk. In the present study, we used blood gas profile to identify the mammary gland health condition in lactating dairy cows.

**Abstract:**

The blood gas profile is a routine method in the rapid disease diagnosis of farm animals, yet its potential in evaluating mammary health status of dairy cows remains to be investigated. This study was conducted to learn the potential of the blood gas parameter regarding the mammary gland health status in lactating dairy cows. Twenty animals were divided into two groups, the H-SCC group (milk SCC > 122 k/mL) and L-SCC group (milk SCC < 73.8 k/mL), to compare blood gas profiles from different blood vessels and to identify the key parameters associated with milk somatic cell count. H-SCC cows are higher in malondialdehyde content, but lower in SOD and T-AOC activities in the milk, compared to the L-SCC group. In terms of blood gas parameters, most differ across the three vessels, including K^+^, CO_2_ pressure, O_2_ pressure, HCO_3_^−^, base excess in the extracellular fluid compartment, and saturation of O_2_. The Pearson correlation analysis showed that oxygen-related variables in the mammary vein, including oxygen concentrations, O_2_ pressure, and saturation of O_2_, are negatively correlated with levels of malondialdehyde, lactate dehydrogenase, and plasmin in the milk. Our study revealed that oxygen-related variables in the mammary vein can be a marker in suggesting mammary-gland health status in high-yielding cows.

## 1. Introduction

The blood gas profiles are an important index that aids clinicians and researchers in differentiating normality and abnormality of parameters, especially for disease diagnoses in various farm animals [1,2,3]. The development of a pen-side blood gas analyzer has facilitated a more accurate approach to assess the blood gas profile [4,5]. In dairy cows, blood gas profile–based references have been established in both adult cows [6,7] and calves [1,8]. Several studies have suggested significant differences in index in blood gas profile between young and adult cattle, such as serum pH, bicarbonate (HCO_3_^−^), and anion gap (AG) [9,10], indicating that the blood gas profile responds to the physiological status alteration of dairy cows.

At the same time, other studies have shown that some indexes, such as anion and carbon dioxide pressure (pCO_2_) and oxygen pressure (pO_2_), can be link to health-status alterations in mammalian animals. For example, the blood anion concentration is associated with metabolic acidosis in postpartum dairy cows [11]. Similarly, pO_2_ in the bloodstream can induce the permeability of the blood vessels [12]. Moreover, the pH and HCO_3_^−^ in the blood were considered to be strongly associated with clinical assessment scores that separate healthy calves from diarrheic ones [13]. Across dairy cows with different physiological stages, early lactating cows, especially high-yielding ones, are suspected to be in a sub-healthy status [14], experiencing symptoms such as mammary gland leakage [15,16] and mammary peroxidation [17], which ultimately lead to mammary dysfunction or even mastitis. However, index evaluation refers to mammary health status, including mammary oxidative stress, mammary permeability, and milk SCC, which largely rely on laboratory facilities. Thus, installing an accurate, facilitated, and real-time analytic approach based on blood gas profile would be a rapid tool to evaluate the health status of mammary gland in high-yielding dairy cows.

To learn the potential of applying blood gas profile as index reflecting mammary gland health status of high yielding dairy cows, we compared blood gas profiles (jugular artery, jugular vein, and mammary vein) between high-yielding dairy cows with high or low SCC in the milk. Moreover, the association analysis between the blood gas profile and health parameter concerning the mammary gland was conducted to identify key gas parameters as potential biomarkers regarding mammary health status in high-yielding dairy cows.

## 2. Materials and Methods

### 2.1. Animals and Management

The experimental protocols were approved by the Animal Care Committee at the Zhejiang University (Hangzhou, China). The animal study was carried out while following Zhejiang University’s guidelines. Based on a power analysis with high reliability (Appendix A), 20 multiparous Holstein dairy cows (means ± standard deviation (SD); parity = 2.7 ± 0.49, DIM = 61.8 ± 6.30) were selected from 70 high-yielding dairy cows at peak stage in Hangzhou Zhengxing Dairy Farm (Hangzhou, China). They were kept in a free-stall barn. All the dairy cows were milked and fed 3 times daily, at 07:00, 13:00, and 19:00, respectively. The cows were fed total mixed ration with an approximate concentrate-to-forage ratio of 55:45 (DM basis) and had free access to water.

### 2.2. Sample Collection and Measurement

Diets were sampled before feedings in the morning and evening of the same days of milk sampling and were then stored at −80 °C till chemical composition analysis. The ingredients and chemical compositions of the total mixed ratio are presented in Appendix A.

Milk yields were determined at a milk-recording facility in the farm (Waikato Milking Systems NZ Ltd., Waikato, Hamilton, New Zealand) for 3 consecutive days. In terms of milk samples, one subsample of the milk was collected with a proportion of 4:3:3 relative to morning, afternoon, and evening milking with a bronopol tablet (milk preservative, D & F Control Systems, San Ramon, CA, USA) and used for infrared analysis of milk components and somatic cells, using a spectrophotometer (Foss-4000, Foss, Hillerød, Denmark), within 2 days of collection. Milk yields were recorded by using milk-sampling devices (Waikato Milking Systems NZ Ltd., Waikato, Hamilton, New Zealand) for 3 consecutive days. For the evaluation of oxidative-stress-related variables, the milk malondialdehyde (MDA) concentration was determined according to Suriyasathaporn et al [18]. In brief, a milk sample was mixed with trichloroacetic acid, and then thiobarbituric acid was added. The prepared mixture was boiled for 30 min and then cooled immediately, and the MDA was determined with spectrophotometer (Leng Guang SFZ1606017568, Shanghai, China). The glutathione peroxidase (GSH-Px) and superoxide dismutase (SOD) were determined by using a commercial kit (Nanjing Jiancheng Bioengineering Institute, Nanjing, China), according to previously reported procedures [19,20], respectively. The total antioxidant capability (T-AOC) was examined by commercial kits (Nanjing Jiancheng Bioengineering Institute, Nanjing City, China). The spectrometric method was applied to evaluate T-AOC. In terms of the variables related with mammary-gland permeability, the levels of Na^+^ and K^+^ in the milk were measured with atomic absorption spectrophotometry (DSI-903B; Shanghai Xunda Medical Instrument Co., Ltd., Shanghai, China), according to previous reported method [21]. The milk bovine serum albumin (BSA) was determined by a radial immunodiffusion assay (Commercial kits provided by Nanjing Jiancheng Bioengineering Institute, Nanjing City, China), according to previous reports [22]. The plasmin activity in milk was determined by following Reference [23]. The activity of lactate dehydrogenase (LDH) in the milk was determined with a colorimetric method, using a commercial kit (Nanjing Jiancheng Bioengineering Institute, Nanjing City) [15].

Blood samples (10 mL) were collected by using two 5 mL pro-coagulation tubes each from the coccygeal artery, coccygeal vein, and mammary vein, respectively. The sample was collected 3 h after morning feeding. One subsample from the three vessels was collected to determine the blood gas profiles immediately, with a portable blood gas parameter analyzer (Abbott, Princeton, NJ, USA). The parameters of blood gas profile included Na^+^, K^+^, total carbon dioxide concentration (TCO_2_), ionized calcium (iCa^2+^), hematocrit (HCT), hemoglobin (HGB), pH, pO_2_, pCO_2_, HCO_3_^−^, base excess in the extracellular fluid (BEecf), and saturation of oxygen, respectively. 

### 2.3. Measurement and Statistical Analysis

The oxygen concentration in the vessel was calculated by using the following equation: concentrations of oxygen = 0.003 × pO_2_ + 1.34 × HGB × saturation of oxygen, which was presented previously [24].

The somatic cell count was transformed by means of a log transformation and used as a response variable. The lactation performance, oxidative-stress-related variables, and oxygen concentration of H-SCC and L-SCC animals were analyzed by using Students’ *t*-test, using SAS software (9.21, SAS Institute, Cary, NC, USA); the differences were evaluated by Duncan’s multiple range tests. The SAS (9.21) MIXED model was applied to analyze gas profiles of blood in the coccygeal artery, coccygeal vein, and mammary vein. Gas profiles of blood in three vessels were analyzed by considering vessel, SCC, and interaction of vessel SCC as fixed effects and the cow as a random effect. Significance was declared at *p* ≤ 0.05, and 0.05 < *p* ≤ 0.10 was considered as a trend. The Pearson correlation between different indexes was conducted by using PROC CORR in SAS (9.21) software, and significance was defined as a *p* < 0.05.

## 3. Results

### 3.1. Lactation Performance

The lactation performances of the H-SCC and L-SCC cows are presented in Table 1. Milk SCCs were significantly higher in H-SCC cows than L-SCC cows (*p* < 0.01). No significant differences were found in the milk yield and the concentrations of milk protein, fat, lactose, or urea nitrogen between animals in the two groups (*p* > 0.05).

### 3.2. Blood Gas Parameter

The blood gas profiles in the coccygeal vein, coccygeal artery, and mammary vein of cows in H-SCC and L-SCC groups are presented in Table 2 and Appendix A. In terms of vessel differences, the concentrations of K^+^ and TCO_2_ are greater in the coccygeal vein and mammary vein compared with the coccygeal artery (*p* < 0.01), respectively. In the meantime, the levels of iCa^2+^ and the pH are higher in the coccygeal artery and mammary vein compared with the coccygeal vein, respectively (*p* < 0.01). Moreover, concentrations of HCO_3_^−^ and BEecf are higher in the mammary vein than those of the coccygeal vein or coccygeal artery, respectively (*p* < 0.01). Finally, a greater pO_2_ and lower pCO_2_ are identified in the coccygeal artery, compared with the coccygeal vein and mammary vein, respectively (*p* < 0.01). Regarding the comparison between H-SCC and L-SCC animals, higher K^+^ was observed in the vessels in H-SCC cows in comparison to L-SCC cows (*p* <0.01). Compared with L-SCC cows, pO_2_ are lower in H-SCC cows (*p* < 0.01). Specifically, pO_2_ in coccygeal artery (*p* = 0.05) and mammary vein (*p* = 0.05) of H-SCC cows are lower than those of L-SCC animals, respectively. The oxygen saturation rate is higher in the artery compared with that in the coccygeal vein and mammary vein, respectively (*p* < 0.01). The oxygen levels in the three vessels of high-yielding dairy cows are presented in Table 3. The oxygen content in the mammary vein is higher in L-SCC-cows in comparison to that of H-SCC ones (*p* < 0.01). However, in the coccygeal artery and coccygeal vein, the oxygen contents are similar between H-SCC- and L-SCC animals (*p* > 0.05).

### 3.3. Oxidative-Stress-Related Variables and Mammary-Gland Permeability

We further determined variables related to tissue permeability and lipid peroxidation status in the milk of these high-yielding dairy cows (Table 4). In terms of oxidative stress status, milk MDA concentrations were lower in L-SCC cows when in compared to those of H-SCC animals (*p* = 0.02). The activities of SOD (*p* < 0.01) and T-AOC (*p* = 0.06) in the milk were higher in L-SCC animals when compared with H-SCC ones, respectively. No difference was identified in milk GSH-Px activity between the two groups (*p* = 0.21). In terms of variables indicating mammary permeability, the Na+ concentration was similar between L-SCC and H-SCC cows (*p* > 0.05), while the K^+^ level tended to be greater in L-SCC animals, as compared to H-SCC cows (*p* = 0.08), and this led to an elevated tendency of the Na^+^/K^+^ ratio in H-SCC cows (*p* = 0.07). In terms of permeability variables, the BSA level was similar between cows in different groups (*p* = 0.25), while the activities of plasmin (*p* = 0.05) and LDH (*p* = 0.02) were lower in L-SCC animals in comparison to the cows in the H-SCC group, respectively.

### 3.4. Correlation Analysis between the Measured Variables

The quantitative correlations between the oxidative stress/permeability-related index in the milk and blood gas profiles in the coccygeal artery (Figure 1A), coccygeal (Figure 1B), and mammary vein (Figure 1C) were calculated. For the coccygeal artery and coccygeal vein, no blood gas index is significantly correlated with either oxidative-stress- or permeability-related parameters in the milk (*p* > 0.05). For the mammary vein, the pO_2_ (R = −0.62, *p* < 0.01), oxygen saturation rate (R = −0.62, *p* < 0.01), and oxygen concentration (R = −0.70, *p* < 0.01) are negatively correlated with milk plasmin concentrations, respectively (Figure 1C and Appendix A). At the same time, pO_2_ (R = −0.74, *p* < 0.01), the oxygen saturation rate (R = −0.78, *p* < 0.01), and oxygen concentration (R = −0.86, *p* < 0.01) in the mammary vein are negatively correlated with milk MDA concentrations, respectively (Figure 1C and Appendix A). Similarly, the pO_2_ (R = −0.78, *p* < 0.01), oxygen saturation rate (R = −0.78, *p* < 0.01), and oxygen concentration (R = −0.78, *p* < 0.01) in the mammary vein are negatively associated with milk LDH activity, respectively (Figure 1C and Appendix A).

## 4. Discussion

During the last few decades, blood-gas-based evaluation has been applied in various aspects of the dairy industry. It is suggested that blood gas profiling in the coccygeal vein is useful for monitoring the acid–base balance in dairy cows [25]. Moreover, blood gas analysis of the coccygeal vein can be a good assessment of subacute ruminal acidosis of dairy cows [26]. In addition, blood gas profiles in the coccygeal vein can be a detailed and accurate diagnosis methodology of health status in neonatal calves [27]. These findings suggest that the blood gas profile can be potentially used as diagnostic tool for health status evaluation. To clarify the potential of blood gas profile in mammary gland metabolic homeostasis of high-yielding dairy cows, we determined blood-gas-related variables in the coccygeal artery, coccygeal vein, and mammary vein of high-yielding dairy cows with high or low milk SCC. It is shown that several parameters, including K^+^, HCT, HGB, pO_2_, and O_2_ saturation, are different between high-yielding cows with different SCCs. We noticed that most of variables differed between cows with different milk SCCs are oxygen-metabolic-related variables. For example, HGB is the main carrier of oxygen for mammalian animals [28]; the higher activity in L-SCC-cows could be one of the causes for the greater available oxygen in the vessels, in relative to H-SCC-cows. It is suggested that lower partial oxygen pressure induced a depressed apoptosis rate in human germ cells [29]. Similarly, Mecklenburgh et al. suggested that the apoptosis rate of blood neutrophils is depressed under hypoxia condition [30]. Other studies suggested that the lower apoptosis rate of neutrophils in milk and blood would lead to an elevated milk SCC of lactating dairy cows [31,32]. Thus, we speculated that lower pO_2_ induced a higher milk SCC for high-yielding dairy cows by delaying the apoptosis rate of neutrophils in the milk and blood.

The higher milk MDA concentration in H-SCC dairy cows suggested that high-yielding cows in H-SCC groups suffered from more severe oxidative stress [18]. Moreover, reduced activities of T-AOC and SOD in the milk indicated a reduced antioxidant capacity in dairy cows with high SCC. Concentrations of Na^+^, K^+^, LDH, and plasmin are considered to be important indexes for mammary permeability evaluation [15,33]. In the current study, higher levels of K^+^, ratio of Na^+^ to K^+^, plasmin, and LDH suggested that the mammary permeability is elevated in H-SCC dairy cows. The more severe oxidative stress and higher permeability suggested that H-SCC cows have a potential health risk linked to their mammary gland. Moreover, we further investigated whether the oxygen status in different vessels is associated with the health status of the mammary gland. However, significant negative associations were observed between pO_2_, oxygen saturation rate, and oxygen concentration in the mammary vein, but not in the coccygeal artery and coccygeal vein, with lipid peroxidation product (MDA) and mammary tight junction leakage (LDH and plasmin). Increased tight junction permeability can be induced when cells are in a low-oxygen environment [34]. Although a negative correlation between mammary permeability and mammary vein oxygen level was identified, the causative relationship between mammary oxygen availability and permeability is yet to be validated. A lipid peroxidation product can be induced by various factors, such as milk SCC [18], tight junction leakage [35], or heat stress [36]. However, our study had the limitation of having SCC levels lower than what mastitis cows would be expected to have. Hence, further studies may emphasize low- and high-SCC (above 200,000 cells/mL) dairy cows. Our study suggested that available oxygen can be another factor regulating lipid peroxidation in mammary gland, and we expect that the blood gas parameter can be potentially utilized as an in-line detection system in the diagnostics of the mammary-gland health of dairy cows.

## 5. Conclusions

In summary, most of the blood gas parameters in the coccygeal artery, coccygeal vein, and mammary vein of high-yielding dairy cows are different from each other. The blood gas index of the three blood vessels, especially ones related to oxygen availability, are lower in H-SCC animals than L-SCC ones. In relation to animals in the L-SCC group, H-SCC animals exhibit more severe oxidative stress and higher permeability in the mammary gland, and these symptoms are associated with their lower oxygen availability in the mammary vein. Our study suggested that data from blood gas parameters in the mammary vein can be used for a potential rapid evaluation of the mammary-gland health status of high-yielding dairy cows. The molecule mechanism of available oxygen in the vessels and it how it relates to mammary-gland regulation remains to be investigated in the future. 

## Figures and Tables

**Figure 1 animals-12-01484-f001:**
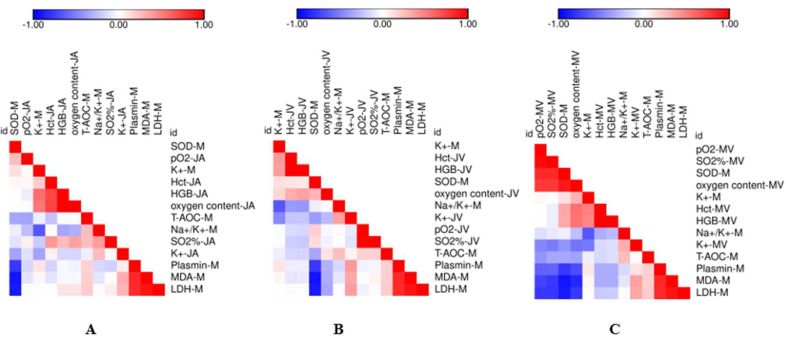
Pearson correlation between oxidative stress/permeability related index in the milk and blood gas profiles in coccygeal artery (**A**), coccygeal vein (**B**), and mammary vein (**C**).

**Table 1 animals-12-01484-t001:** Lactation performance of high-yielding dairy cows with different somatic cell count (SCC).

Item ^1^	L-SCC	H-SCC	SEM	*p*-Value
Milk yield, kg/d	56.1	55.8	1.24	0.62
Milk composition, g/100 g				
Protein	3.10	2.97	0.059	0.13
Fat	3.80	3.83	0.245	0.92
Lactose	4.85	4.93	0.134	0.67
Urea nitrogen, mg/dL	13.8	14.1	0.51	0.61

^1^ H-SCC: lactating dairy cows with higher milk SCC. L-SCC: lactating dairy cows with lower milk SCC.

**Table 2 animals-12-01484-t002:** Blood gas profile comparison in coccygeal artery, coccygeal vein, and mammary vein in high-yielding dairy cows with high and low somatic cell counts (SCCs).

Item ^1^	L-SCC			H-SCC			SEM	*p*-Value		
	Coccygeal Artery	Coccygeal Vein	Mammary Vein	Coccygeal Artery	Coccygeal Vein	Mammary Vein		Vessel	SCC	Interaction
Na^+^, mmol/L	136.9	136.5	137.2	136.3	135.9	136.9	0.30	0.26	0.24	0.94
K^+^, mmol/L	3.76 ^b^	4.03 ^a^	3.65 ^b^	3.87 ^b^	4.23 ^a^	3.79 ^b^	0.0390	<0.01	<0.01	0.80
TCO_2_, mmol/L	30.9 ^b^	31.7 ^b^	32.5 ^a^	30.8 ^b^	31.2 ^b^	33.2 ^a^	0.33	<0.01	0.94	0.56
iCa, mmol/L	1.19 ^b^	1.20 ^a^	1.14 ^b^	1.19 ^b^	1.23 ^a^	1.15 ^b^	0.101	<0.01	0.40	0.74
HCT	0.245	0.236	0.236	0.244	0.234	0.232	0.006	0.62	0.07	0.51
HGB, g/dL	8.30	8.33	8.34	8.10	7.97	7.88	0.115	0.91	0.04	0.79
pH	7.49 ^a^	7.45 ^b^	7.51 ^a^	7.50 ^a^	7.43 ^b^	7.50 ^a^	0.009	<0.01	0.57	0.60
pCO_2_, mmHg	37.6	45.3 ^a^	40.7 ^b^	35.9	44.8 ^a^	41.2 ^b^	0.45	<0.01	0.35	0.35
pO_2_, mmHg	107 ^a^	31.2	40.3	101 ^b^	31.1	33.6	1.40	<0.01	0.03	0.29
HCO_3_^−^, mmol/L	29.8 ^b^	30.4 ^b^	31.3 ^a^	29.6 ^b^	29.8 ^b^	31.9 ^a^	0.31	<0.01	0.85	0.53
BEecf, mmol/L	6.70 ^b^	5.60	7.90 ^a^	7.00 ^b^	5.70	8.70 ^a^	0.446	<0.01	0.28	0.72
Saturation of O_2_ %	98.2 ^a^	60.0	77.8 ^b^	98.7 ^a^	60.3	69.3	2.13	<0.01	0.15	0.05

^1^ TCO_2_ = total carbon dioxide concentration; iCa^2+^ = ionized calcium, HCT = hematocrit, HGB = hemoglobin, pH, pO_2_ = oxygen pressure, pCO_2_ = pressure of carbon dioxide, BEecf = base excess in the extracellular fluid. H-SCC: lactating dairy cows with higher milk SCC. L-SCC: lactating dairy cows with lower milk SCC. ^a,b^ Means in the same row with different superscripts are statistical different (*p* < 0.05).

**Table 3 animals-12-01484-t003:** Oxygen concentrations in different vessels of high-yielding dairy cows with different somatic cell counts (SCCs).

Item ^1^	L-SCC	H-SCC	SEM	*p*-Value
Coccygeal artery	11.4	11.2	0.27	0.54
Coccygeal vein	10.6	11.1	0.54	0.56
Mammary vein	9.15 ^a^	7.72 ^b^	0.224	<0.01

^a,b^ Means in the same row with different superscripts are statistical different (*p* < 0.05). ^1^ H-SCC: lactating dairy cows with higher milk SCC. L-SCC: lactating dairy cows with lower milk SCC.

**Table 4 animals-12-01484-t004:** Milk variables related to oxidative stress status and tight junction integrity of mammary gland of high-yielding dairy cows with different somatic cell counts (SCCs).

Item ^1^	L-SCC	H-SCC	SEM	*p*-Value
Oxidative-stress-related variables				
MDA, nmol/L	9.52 ^b^	10.45 ^a^	0.26	0.02
GSH-Px, U/mL	138.1	128.3	5.39	0.21
SOD, U/mL	119.0 ^a^	104.8 ^b^	2.76	<0.01
T-AOC, U/mL	14.0	11.9	0.73	0.06
Mammary-gland permeability				
Na^+^, mmol/L	19.2	20.6	0.71	0.21
K^+^, mmol/L	37.8	35.7	0.81	0.08
Na^+^/K^+^	0.51	0.57	0.024	0.07
BSA, g/L	0.72	0.78	0.034	0.25
Plasmin, U/L	2.58 ^b^	3.17 ^a^	0.21	0.05
LDH, U/L	223 ^b^	254 ^a^	8.5	0.02

^a,b^ Means in the same row with different superscripts are statistical different (*p* ≤ 0.05). ^1^ MDA, malonaldehyde; GSH-Px, glutathione peroxidase; SOD, superoxide dismutase; T-AOC, total antioxidant capacity; BSA, bovine serum albumin; LDH, lactate dehydrogenase; H-SCC, lactating dairy cows with higher milk SCC; L-SCC, lactating dairy cows with lower milk SCC.

## Data Availability

Not applicable.

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
