# Peer review of "Multiple-Vessel-Based Blood Gas Profiles Analysis Revealed the Potential of Blood Oxygen in Mammary Vein as Indicator of Mammary Gland Health Risk of High-Yielding Dairy Cows"

_animals, 2022, doi:10.3390/ani12121484_

Round 1

Reviewer 1 Report

English: This is of a poor standard which makes the paper difficult to read and understand, and also results in some apparent errors of science. I recommend that after this first review round that the paper is edited either by a native English speaker who is familiar with the subject or by one of the many available editing services before it is submitted for further review

Title: "the potential of blood oxygen as indicator of mammary gland 
health risk of high-yielding dairy cows" This is not risk as you have no future data you just have what is happening now, so you have blood oxygen measurement from the mammary vein as an alternative to measuring SCC. This is not a sensible test when SCC is so simple and non-invasive. measuring blood gases in high and low cell count cows is interesting, using those gases to predict SCC is not

Statistical analysis

I couldn’t work out what was done, this section needs extensive rewriting

Some examples:

student t-test for cell count given lactation performance. What is lactation performance? It’s categorical as you use a t-test. What categories

for the mixed model you don’t state clearly what are the outcome variables and what are your predictor variables. Did you use a selection procedure if so what? Also how did you ensure that the data met the requirements of your model

The reader should be able to repeat the process you used (as in a recipe for making a cake). You have just given me a list of ingredients

You state you  analysed the “effects of …”. You didn’t you only looked at associations

There is no such thing as a statistical trend (Nead et al 2018, Gibbs and Gibbs 2015) – you don’t describe results between 0.03 and 0.05 as trending towards non-significance, so 0.05 to 0.10 is not trending towards significance. There is nothing wrong in an exploratory study, where you can make no conclusions  form your data and analysis about pathology pathways, utility, or predictive ability, setting a p-value of 0.10 to mean data that needs further investigation. Just have courage and use that and except that as all of your conclusions need confirmation there is nothing wrong with a high threshold.

3.1 lactation performance

This is almost entirely methods not results.

122 is not very high SCC

3.2 blood gas

There is no such thing as a jugular artery

Where possible please state actual p-values not arbitrary thresholds

Discussion

Much of the first paragraph is repetition of what is  in the introduction and what isn’t should be

“The elevated milk SCC can be induced by multiple factors, such as oxidative stress  and mammary permeability” This is an example of poor science from poor English. Mammary permeability does not induce SCC, it might allow it (increasing cell passage). In contrast oxidative stress may be a factor that increases mastitis risk but it does not induce SCC directly and in relation to increased SCC acts in a completely different way from permeability

“Increased tight junction permeability can be induced when cells are in a low oxygen environment“ – are your 02 levels in the mammary vein low enough for you to conclude that mammary tissue is in a low 02 environment.

“Lipid peroxidation product can be induced by various factors, such as milk SCC [26]”. No the reference you cite simply shows that SCC and MDA go up together not that MDA is caused by SCC. It’s actually quite likely that peroxidation products cause SCC increase

Reviewer 2 Report

The manuscript entitled “Multiple-vessel-based blood gas profiles analysis revealed the potential of blood oxygen as indicator of mammary gland health risk of high-yielding dairy cows” was reviewed. The content of this study is of clinical significance and could be used to support diagnostic and consequently therapeutic decisions made by farm animal practitioners. The quality of presentation is adequate and can be improved. More specifically the following issues should be taken into consideration

Title

The title reflects the content of the manuscript. Maybe it would be more appropriate to write mammary vein blood gas profile instead of multiple vessel so as to highlight that mammary vein blood was the most appropriate

Introduction

The introduction provides sufficient background and includes all relevant references.

Materials and Methods

Although the research design is appropriate and the methods are adequately described, the following corrections should be made.

Lines 80-86: please provide further details regarding the sampling procedure for the milk samples used for biochemical parameters. As presented, it is unclear whether milk samples were collected within one day or for 3 consecutive days, as referred for milk yield recordings. In addition, were they composite milk samples? The aliquots were mixed before analysis or each aliquot was analyzed separately?

Lines 113-114: Which parameters were analyzed in blood serum?

Results

Lines 132-142: Some of this information such as the criteria for the selection of the 20 animals, the separation of the animals in two groups according to SCC which are important for the understanding of your study design should be referred at the Materials and Methods section. In addition, since SCC was the mail criterion for the group allocation of the animals, the comparison of SCC between groups is not necessary.

How were these cut-off points for SCC were selected?

Discussion

Line 224: you had significantly higher concentration of plasmin and higher LDH activity and numerically lower K and Na:K ratio. Please correct it

Conclusions

Although it is already referred at the first sentence, it has to be highlighted here that only mammary vein blood gas analysis is useful for the determination of mammary health

Line 267: in relation to
